# Community perception differences regarding ecosystem goods and services in reservoir landscapes with and without inter-basin water transfer: Implications for the Sustainable Development Goals

Lucianna Marques Rocha Ferreira[1]*, Franciely Ferreira Paiva[2],
Maria Eduarda Santana Veríssimo[2], Lívia Maria Osório de Sousa[2],
Evaldo de Lira Azevêdo[3], José Etham de Lucena Barbosa[4], Joseline Molozzi[4]

1 Laboratório de Ecologia de Bentos, Universidade Estadual da Paraíba - UEPB, Campina Grande, Paraíba, Brazil, 2 Programa de Pós-graduação em Ecologia e Conservação, Universidade Estadual da Paraíba - UEPB, Campina Grande, Paraíba, Brazil, 3 Instituto Federal de Educação, Ciência e Tecnologia da Paraíba, Princesa Isabel, Paraíba, Brazil, 4 Departamento de Biologia, Universidade Estadual da Paraíba - UEPB, Campina Grande, Paraíba, Brazil

* lucianna.mrf@gmail.com

## Abstract

Water is a fundamental ecosystem good and service (EGS) for supporting life on Earth. In arid and semiarid regions, water scarcity is a recurring problem that limits socioeconomic activities and the achievement of Sustainable Development Goals (SDGs). Inter-basin water transfer (IBWT) schemes have been employed to alleviate the impact of water scarcity. This study explored how IBWT affects the perceptions of riverside communities regarding EGSs within a semiarid reservoir landscape, assessing the interplay between perceived EGSs, SDGs, and land use and land cover (LULC). Furthermore, this study evaluated the influence of sociodemographic factors on these perceptions. The study was conducted across eight reservoir landscapes, with four reservoirs receiving and four not receiving IBWT. Semi-structured forms and participatory mapping were used to discern and map the EGSs as perceived by the communities. These perceived EGSs were then linked to the SDGs. The communities identified 29 EGSs classes (provision, regulation and maintenance, and cultural services) in the set of reservoir landscapes studied. Provision services were the most frequently mentioned (78.53%). It was found that educational level significantly influenced community perceptions of EGSs (p = 0.003). Particularly, provisioning services associated with the LULC water were mentioned more frequently than other LULC types (p = 0.02). Forest formations were the primary providers of regulation and maintenance services compared to water bodies, land use mosaics, and floodplains ($p_{adjusted}$ = 0.02) and received more citations for cultural services than land use mosaics and built-up areas ($p_{adjusted}$ = 0.02). Cultural services were predominantly

**Data availability statement:** All relevant data are within the article and its supporting information files.

**Funding:** The work was supported by the Fundação de Apoio à Pesquisa do Estado da Paraíba (FAPESQ) (https://fapesq.rpp.br/), under grant number 403/2021, awarded to JELB; and by the Conselho Nacional de Desenvolvimento Científico e Tecnológico (CNPq) (https://www.gov.br/cnpq/pt-br), under grant numbers 3149/2021, 409348/2022-8, and 428602/2018-5, all awarded to JM. The funders had no role in study design, data collection and analysis, decision to publish, or preparation of the manuscript.

**Competing interests:** The authors have declared that no competing interests exist.

acknowledged by individuals residing near reservoirs that received IBWT (p = 0.006), while those near non-IBWT reservoirs more often reported regulation and maintenance services (p = 0.003). Provisioning services were strongly linked to the SDGs (p = 0.0001) and can substantially facilitate SDGs attainment, notably impacting goals 1, 2, 3, 12, and 15. The presence of IBWT significantly shapes community perceptions of reservoir landscape elements in the semiarid region.

## Introduction

The 2030 Agenda for Sustainable Development was formulated in response to various global challenges, encompassing environmental issues such as habitat and biodiversity loss, biological invasion, and climate change, as well as socioeconomic concerns such as poverty, hunger, social inequality, and political conflict. This comprehensive framework, aimed at promoting global peace, prosperity, and the eradication of hunger, outlines 169 targets across 17 Sustainable Development Goals (SDGs) adopted by the United Nations in 2015, with the endorsement of 195 signatory countries [1,2].

To achieve SDGs, socioeconomic initiatives must consider biotic and abiotic factors and their interactions within the landscape [3]. This approach is vital because the structure and function of ecosystems provide benefits that can drive improvements in human well-being, collectively referred to as ecosystem goods and services (EGSs) [4]. EGSs are classified into three primary types: provisioning, which includes essentials like drinking water and food; regulation and maintenance, which encompasses services such as climate regulation; and cultural, which covers cultural and recreational activities [5]. These services are indispensable for meeting basic human needs and enhancing overall well-being [3,6,7].

In the face of global challenges, the crises and conflicts related to water resources stand out, particularly in arid and semiarid regions. Converting forested areas into built environments, coupled with the impacts of climate change, further exacerbates water insecurity [8]. In response to the looming threat of water scarcity, reservoir construction has been adopted to support and sustain human settlements [9]. Reservoir landscapes provide several EGSs, such as water, fish stocks, and recreational opportunities, and play a critical role in the pursuit of sustainable development [6]. Although SDGs do not explicitly mention reservoir landscapes in their targets, it is known that many ecosystem services (ESs) provided by these landscapes are essential for achieving the goals of several SDGs [1]. This is because most EGSs support the attainment of multiple SDG targets [1,7,10,11]. For example, the provision and enhanced production of plant-based foods not only support SDG 1 (No poverty) and SDG 2 (Zero hunger), but also contribute to SDG 13 (Climate action) by promoting sustainable agricultural practices [11].

However, the construction of reservoirs does not guarantee water security, as they are vulnerable to extreme natural events, such as prolonged droughts and pollution, which compromise both water availability and quality [9]. Consequently, inter-basin

water transfer (IBWT) projects have been proposed as strategic solutions to address the challenges posed by water insecurity, including shortage of human water supply, by connecting river basins to areas with water demand [12,13]. Specifically, IBWT refers to the diversion of a river from one watershed to another [12]. This alteration in the course of the river, redirected to supply reservoirs, transforms previously intermittent rivers into perennial ones, leading to changes in the landscape and the biological composition of ecosystems [14]. Moreover, the availability of EGSs is significantly influenced by LULC changes brought about by the implementation of IBWT projects [15]. Water-related ecosystem services increase due to the enhanced water volume resulting from IBWT [15]. However, forest ecosystem services tend to decline as forested areas are often lost and replaced by agricultural land [15]. Agriculture, in particular, is one of the LULC types that benefits most from IBWT projects [13].

IBWT projects have direct socioeconomic impacts, as highlighted by [13]. These authors analyzed the socioeconomic effects of the São Francisco River Integration Project in Brazil, which aimed to strategically transfer water from the São Francisco River to reservoirs in other watersheds in the semi-arid Northeast region of the country. Through future projection models, specifically for 2035, they observed increases in productivity, Gross Domestic Product (GDP), employment, and household consumption, particularly in rural areas where agriculture is predominant. The enhanced water availability and food production resulting from IBWT projects directly contribute to reducing poverty, mitigating social inequality, and improving public health [13]. Although the authors do not explicitly link their findings to EGSs and the goals of the 2030 Agenda, it is evident that the São Francisco River Transposition Project enhances the ecosystem services provided by water, contributing to the achievement of SDGs 1 (No Poverty), 2 (Zero Hunger), 6 (Clean Water and Sanitation), 8 (Decent Work and Economic Growth), 10 (Reduced Inequalities), 12 (Responsible Consumption and Production), 13 (Climate Action), and 15 (Life on Land).

Given the above, understanding local community perceptions of a landscape is essential for identifying EGSs, as this can significantly contribute to achieving the SDGs [1,3]. The perceptions of riparian communities highlight the essential landscape and ecosystem benefits that are crucial for sustaining life, including water and nutrients, cultural and educational values, and personal environmental experiences [7]. It is also important to note that demographic factors, such as ethnicity, gender, age, occupation, family income, and education level, play a crucial role in shaping individual and collective understanding of reservoir landscape elements, influencing the identification and valuation of EGSs [6].

This study is, to our knowledge, the first to examine how the coexistence of riverside communities with IBWT impacts the perception of EGSs within the reservoir landscape of the semiarid region while assessing the interactions between perceived EGSs, LULC, and SDGs. The study tested the following hypotheses: *i* Sociodemographic factors significantly influence the perception of EGSs, with notable differences between individuals residing near reservoirs receiving IBWT and those that do not; *ii* Communities living in proximity to reservoirs receiving IBWT report a higher diversity and quantity of ESGs per class and section; *iii* Populations near IBWT reservoirs are more aware the EGSs associated with "water" and "forest formation" LULC types; and *iv* Among the perceived services, provisioning services are most acknowledged by the populations, playing a crucial role in fulfilling SDGs, irrespective of their proximity to reservoirs with or without IBWT.

The novelty of this study lies in the integration of EGS mapping based on the perceptions of the riverside communities and the LULC of the reservoir landscape that receives and does not receive IBWT. Understanding the differences and similarities in EGS perceptions between these two groups has the potential to inform political and environmental management strategies that can effectively contribute to achieving the SDGs. Additionally, this approach can serve as a valuable global reference for semi-arid regions.

## Materials and methods

### Study area

The study was carried out within a reservoir landscape in the semiarid region of Brazil, focusing on the Paraíba River Basin – encompassing the Argemiro de Figueiredo, Camalaú, Cordeiro, Epitácio Pessoa, Poções, Sumé, and Taperoá

reservoirs – and the Araçagi reservoir in the Mamanguape River Basin (Fig 1). These selected reservoirs have been established to fulfill various needs, including human and animal water supply, and to support a wide range of economic activities in the area, such as agriculture, livestock farming, aquaculture, industrial operations, and recreational and tourism ventures [16].

The semiarid region of Brazil is experiencing water crises and the collapse of freshwater ecosystems. Thus, policy decisions have shifted toward implementing the IBWT project in the northeast region of Brazil [17]. In 2005, an installation license was granted for the São Francisco River Integration Project, with the Northern Northeast Hydrographic Basins covering the north and east axes [17]. The northern axis facilitates the transfer of water from the São Francisco River Basin to the Jaguaribe River Basin in Ceará, the Piranhas-Açú River Basin and the Apodi-Mossoró River Basin in Rio Grande do Norte. The east axis directs water from the São Francisco River to the Paraíba River Basin [18].

The first stage of construction along the east axis commenced operations in 2017, channeling water to the São José I, Poções, Camalaú, and Epitácio Pessoa reservoirs and extending towards the Argemiro de Figueiredo reservoir [21]. The

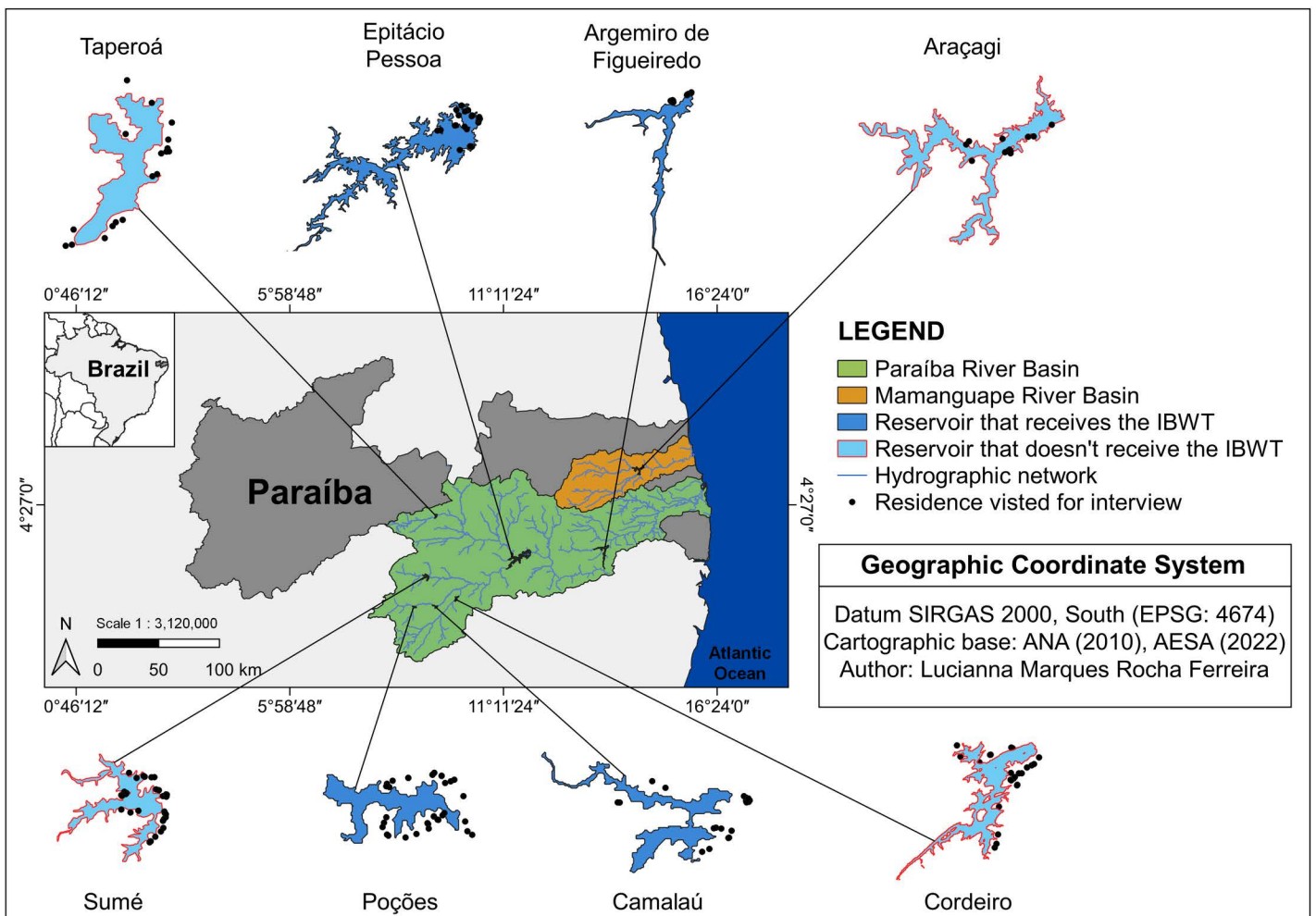

**Fig 1. Location of the investigated reservoirs in Paraíba and the location of the residence visited where the interviews took place, Brazil. Author: Lucianna Marques Rocha Ferreira. References used in the cartographic base**: [19,20].

 

inaugural section of the Acauã-Araçagi Canal was inaugurated in May 2022, facilitating the integration of the São Francisco River Basin and Paraíba River Basin with the Mamanguape River Basin for the Araçagi Reservoir.

It is worth mentioning that the Brazilian semiarid region experienced the worst drought in the last 50 years between 2012 and 2017 [22,23]. The implementation of the IBWT project in 2017 [21] and the increase in rainfall in 2018 [22] gradually impacted the monthly water volumes of the Argemiro de Figueiredo, Camalaú, Epitácio Pessoa, and Poções reservoirs [21,24,25]. For example, the Poções reservoir, the first to receive water from the transposition in March 2017 [21], had a volume of only 3.66% of its maximum capacity in January 2017, placing it in a critical condition (less than 5% of its maximum storage capacity) [25]. However, after the São Francisco River transposition, in January 2022, its capacity had increased to over 20% (29.18% of its maximum capacity), a level considered normal by the Executive Agency for Water Management of the State of Paraíba [25]. The reservoirs receiving water from the São Francisco River transposition are less vulnerable to reaching critical levels of volume and water quality, except in cases where structural failures caused interruptions in the transposition [21]. Conversely, reservoirs such as Araçagi, Cordeiro, Sumé, and Taperoá, which are not benefited by the IBWT project, have water volumes that fluctuate according to climatic phenomena in the Brazilian semiarid region.

According to the Köppen-Geiger classification [26], the Camalaú, Cordeiro, Epitácio Pessoa, Sumé, and Taperoá reservoirs have a BSh (dry semiarid) climate[26]. The dry season in these areas extends for 9–10 months, with an average annual rainfall ranging from 350 to 600 mm [27]. Temperatures vary, reaching monthly lows of 18–22 °C during July and August and monthly highs of 28–31 °C in November and December [27]. The dominant vegetation is hyperxerophilic Caatinga and deciduous and subdeciduous forests [28].

The Argemiro de Figueiredo and Araçagi reservoirs are characterized by an As' climate, indicative of a tropical zone with a dry summer, according to the Köppen-Geiger classification [29]. The average annual air temperature is approximately 26 °C, with an average annual rainfall of around 800 mm in the Argemiro Figueiredo reservoir and 1,130 mm in the Araçagi reservoir [29]. Rain is most abundant from February to July, with the dry season from October to December [27]. The surrounding vegetation includes deciduous and subdeciduous forests [28].

## Sample design

Data on EGSs within the reservoir landscape were obtained from the perceptions of riverside communities near reservoirs, distinguished by their status as receiving or not receiving IBWT. Semi-structured forms, participatory mapping, and LULC classification were used to collect the data. Notably, during the interview period of 2021–2022, only the Argemiro de Figueiredo, Epitácio Pessoa, Camalaú, and Poções reservoirs received IBWT from the São Francisco River. First, a semi-structured forms was used to collect sociodemographic information from participants. Next, participatory mapping was employed, allowing interviewees to pinpoint the locations where the EGSs were offered on the map. Subsequently, LULC classification of each reservoir's landscape was performed to establish the geospatial association between the EGSs identified by the participants during the participatory mapping process and the LULC (Fig 2).

Interviews were conducted in the vicinity of the reservoirs using a one-to-one, door-to-door approach, with inclusion criteria set to living within 200 m of a reservoir, voluntary participation in the research, and being at least 18 years old. We considered a 200 m radius from the reservoir as a criterion based on the premise that closer proximity facilitates greater interaction between people and landscape elements [30]. Google Earth Pro was used to identify potential interviewee residences and all households within this boundary were visited. Acknowledging the unique experiences and relationships individuals have with the reservoir landscape, in some cases, more than one family member was interviewed, with a five-meter distance maintained between participants to minimize interview disruption.

This study was approved by the Ethics Committee of the Universidade Estadual da Paraíba (CEP-UEPB, number: 505383) on October 22, 2021. After approval by the Ethics Committee, we began recruitment on December 13, 2021, and completed it on January 26, 2022. After being informed about the purpose of this research and understanding the

explanations provided in this Informed Consent Form, the participant authorized their participation in the study and gave permission for the data obtained to be used solely for scientific purposes, with their identity being preserved. All participants signed the form in writing, along with the researcher and two witnesses, in two copies of identical content, with one copy remaining with the participant and the other with the researcher.

### Semi-structured form: sociodemographic data

After obtaining the participant's consent to participate in the study, we used semi-structured forms to collect data on gender (male or female), age (in years), education level (ranging from illiterate to postgraduate), occupation, and monthly family income. We considered sociodemographic indicators commonly evaluated to examine differences in community perceptions of EGSs [6].

Monthly family income was categorized into brackets: below the minimum wage, 1–2 minimum wages, 3–4 minimum wages, 5–6 minimum wages, variable, or unspecified. The minimum monthly wage in Brazil in 2021 was R$ 1,110.00, which is equivalent to US$ 205.75 when converted using an average exchange rate of R$ 5.3950 [27].

### Participatory mapping: identification of EGSs and their association with SDGs

EGSs within the reservoir landscape were identified by interviewees using participatory mapping, a method that leverages the specific and contextual knowledge of individuals in a geographic area based on their personal experiences [31]. This

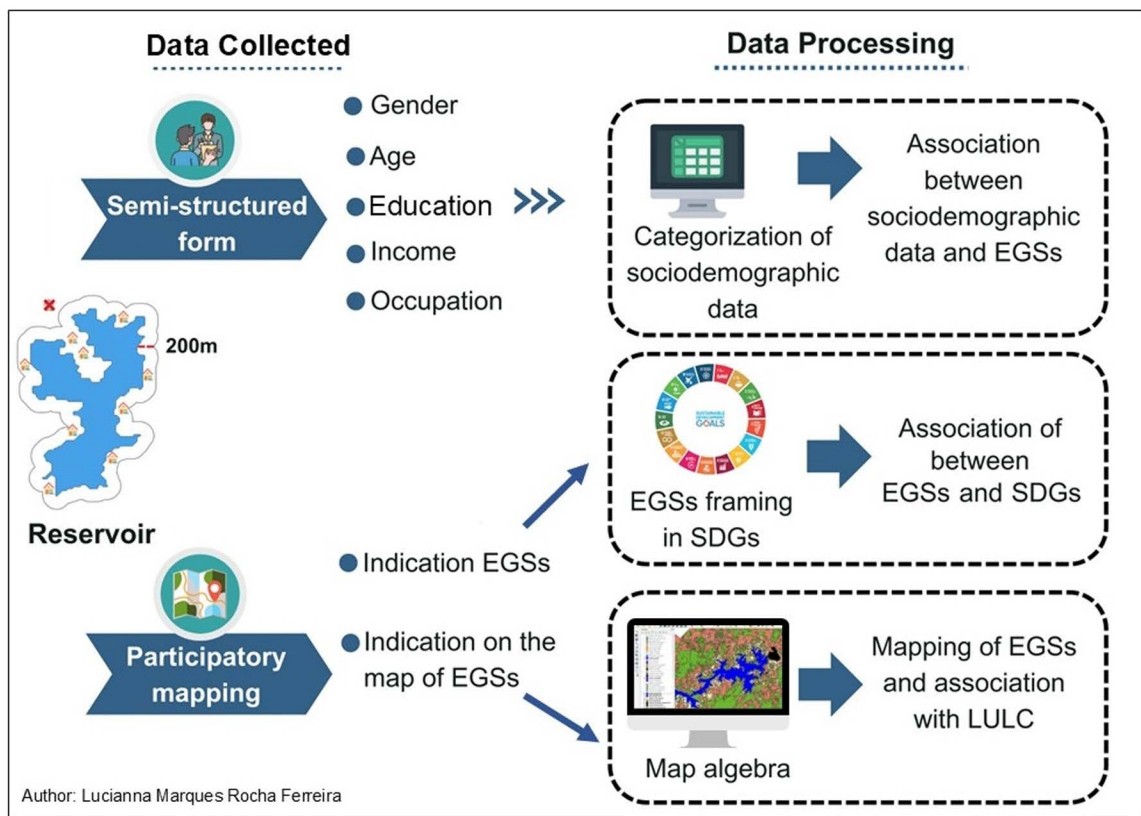

**Fig 2. Scheme for obtaining and processing primary data. Figure created by the author: Lucianna Marques Rocha Ferreira, using Canva Pro (www.canva.com).**

method, particularly relevant to ecosystem services research, involves participants indicating specific locations on a map that provide particular ecosystem goods or services [32].

During the interviews, the participants were presented with an A4-sized aerial image map of the reservoir landscape. After explaining the concept of EGSs and ensuring comprehension, the interviewees were asked to describe the ecosystem benefits provided by the landscape and to pinpoint the locations on the map where these EGSs were available. To this end, they were invited to answer the following questions: "What ecosystem benefits does this landscape provide to people?" and "Indicate on the map where these ecosystem benefits are found."

Responses were recorded and categorized according to the Common International Classification of Ecosystem Services (CICES), version 5.1 [5], as detailed in Table A in S1 Table. This framework organizes EGSs provided by biota, the physical environment, and ecosystems into a structured hierarchy to avoid double counting. The CICES framework categorizes EGSs into three broad sections – provisioning services, regulation and maintenance, and cultural – and further subdivides them into divisions, groups, classes, and class types [5].

Following the identification of EGSs, we linked the specific classes of services mentioned by participants to the 17 SDGs and their 169 targets, established by the United Nations in 2015 [2], as outlined in Table A in S1 Table. This linkage was conducted according to research developed by [7], who used an anonymous online survey to explore connections between SDG targets with environmental ties and EGSs. Our study adapted this approach to accommodate differences in classification between The Economics of Ecosystems & Biodiversity Report for Business [33] framework used by [7] and the CICES framework, establishing equivalencies between the two systems.

To assess the relationship between SDGs and perceived EGSs by communities adjacent to reservoirs with and without IBWT, we focused on three SDG domains: securing basic material needs, enhancing overall human well-being, and fostering sustainable governance policies, according to [1]. This comprehensive approach allowed for a detailed understanding of how local perceptions of ecosystem services can contribute to broader sustainability objectives.

## Land use and land cover classification: mapping of EGSs and their association with LULC

The land use and land cover (LULC) classification of the reservoir landscape was critical for translating information about EGSs, as identified by participants on physical maps through participatory mapping, into a digital format. Therefore, remote sensing and geographic information system (GIS) techniques were used to classify the LULC of each reservoir landscape.

Thematic LULC maps were created using a semi-supervised classification method applied to images acquired by the Sentinel-2 satellite MultiSpectral Instrument (MSI), Harmonized Level-2A. The Sentinel-2 satellite is part of the European Space Agency's (ESA) Copernicus program. ESA provides free access to images of the Earth's surface captured by the sensors on board the Sentinels satellites, and these images have been included in the catalog of the Google Earth Engine (GEE) geospatial platform. All the images used in this study were obtained directly from the public data catalog of the GEE geospatial platform [34].

The classification and accuracy assessment of these thematic maps were conducted using GEE [35]. To compile the mosaic image of each reservoir landscape, the median of each pixel was calculated from satellite images acquired through GEE from November 1, 2021, to December 31, 2022. Only the images with a maximum cloud cover of 10% were selected for this analysis.

We used spectral bands B11, B8, and B4 along with geographic coordinate data obtained in the field using a Garmin 64S GPS as references for creating training and validation areas essential for the semi-supervised classification process. Accordingly, 70% of the designated points for semi-supervised classification was randomly assigned (9999 permutations) as training areas, while the remaining 30% served as validation areas. LULC was classified into the following categories based on field observations: water (river courses and reservoirs resulting from large and small-scale surface dams), aquaculture (fish or shrimp farming), floodplains (areas adjacent to a water that are periodically

flooded), forest formation (tree and shrub vegetation), macrophytes (aquatic plants on the reservoir surface), land use mosaics (agriculture and/or livestock), bare land (area without vegetation), and built-up areas (houses, buildings, and paved roads).

The accuracy of the semi-supervised classification was evaluated using the Kappa Index [36], which measures the agreement between the training and validation areas. Kappa Index (K) values range from zero (indicating poor agreement) to one (indicating excellent agreement) [36,37]. The accuracy assessment followed the criteria established by [37] to evaluate the degree of agreement of thematic classification.

The results showed that all thematic LULC maps produced through semi-supervised classification showed excellent agreement, with Kappa values close to one for all assessed reservoir landscapes: Araçagi (K = 0.996), Argemiro de Figueiredo (K = 0.993), Camalaú (K = 0.996), Cordeiro (K = 0.992), Epitácio Pessoa (K = 0.997), Poções (K = 0.999), Sumé (K = 0.995), and Taperoá (K = 0.997). After verifying the accuracy, the image mosaics for each reservoir landscape were georeferenced to SIRGAS 2000, South (European Petroleum Survey Group: 4674). Thematic LULC maps were created using QGIS version 3.10.7 and GRASS 7.8.3.

Subsequently, a shapefile-type information storage format was established for each LULC class within each reservoir landscape. This allowed the creation of a shapefile for each EGS class associated with LULC, with the number of EGS classes added to the attribute table. For example, agricultural activities were linked to the land use mosaic shapefile, while fishing activities were associated with the water shapefile. All shapefiles were converted to raster format with a pixel resolution of 10m x 10m. Map algebraic techniques were applied to interpolate the EGS-class data for each reservoir landscape. Finally, all maps produced by the respondents were aggregated using the EGSs class variable registered in each pixel, resulting in a thematic map that indicated the number of EGSs per reservoir landscape.

## Statistical analysis

All research data were analyzed using nonparametric statistical tests because the data did not conform to a normal distribution even after transformation attempts. To explore the interactions between sociodemographic variables and the IBWT variable in terms of the number of EGSs and the number of EGSs per section, we employed univariate Permutational Multivariate Analysis of Variance (PERMANOVA) with Euclidean distance and 9999 permutations [38]. We then assessed whether individuals living near reservoirs receiving IBWT perceived a greater number of EGSs, as well as a higher number of EGSs per section, by performing a Mann-Whitney test [39] and PERMANOVA with Euclidean distances and 9999 permutations, respectively. We further examined whether populations adjacent to reservoirs receiving IBWT noted a greater diversity and quantity of EGSs and more EGSs per section for each LULC type (water, aquaculture, floodplains, forest formation, land use mosaic, bare land, and built-up area), employing univariate PERMANOVA test that focused on one factor at a time.

Finally, the contribution of provisioning services perceived by the population towards achieving the SDGs was evaluated using the PERMANOVA test with Euclidean distance and 9999 permutations. All statistical tests were performed using RStudio software, version 4.1.1 [40] with a significance level of 5% or less. The 'vegan' package was used to perform PERMANOVA. For significant interactions, only pairwise post-hoc tests with Euclidean distance and Bonferroni correction were performed to determine the differences between levels of the independent variables using the 'pairwiseAdonis' package.

## Results

### Sociodemographic profile

We interviewed 205 individuals, with the number of participants varying by reservoir landscape due to differences in population density and the acceptance to participate in the research. Specifically, 13 were interviewed around the Araçagi reservoir, 17 at Argemiro de Figueiredo, 18 near Taperoá, 25 at Cordeiro, 27 at Camalaú, 33 at Poções, 35 in the community

near Sumé, and 37 at Epitácio Pessoa. Of these, 55.61% (N = 114) live near reservoirs that receive IBWT (Argemiro de Figueiredo, Epitácio Pessoa, Camalaú, and Poções), while 44.39% (N = 91) reside close to reservoirs that do not (Araçagi, Cordeiro, Sumé, and Taperoá).

A slight majority of the interviewees were women (54.15%, N = 111), with 32.20% (N = 66) near reservoirs receiving IBWT and 21.95% (N = 45) near those that do not. The gender distribution was relatively balanced, with 45.84% (N = 94) identifying as male (23.41%, N = 48, near reservoirs with IBWT, and 22.44%, N = 46, near those without, Table B in S1 Table). Ages ranged from 18 to 85 years (Mean = 47.81 and SD ± 16.14), with the largest group aged 31–45 (31.22%, N = 64), including 16.59% (N = 34) living close to reservoirs with IBWT and 14.63% (N = 30) near those without.

Education levels varied: only 4.39% (N = 9) held a university degree (1.47%, N = 3 near reservoirs with IBWT and 2.93%, N = 6 near reservoirs without). A total of 15.61% (N = 32) reported being illiterate, with 9.27% (N = 19) near reservoirs with IBWT and 6.34% (N = 13) near those without. Most participants (47.80%, N = 130) had not completed elementary school, including 36.59% (N = 75) near reservoirs with IBWT and 32.83% (N = 55) near those without (Table B in S1 Table).

Regarding income, 58.05% (N = 119) of the respondents reported earning one to two minimum wages (reservoirs with IBWT: 31.22%, N = 64; without IBWT: 26.83%, N = 55), and 34.63% (N = 71) earned less than one minimum wage (reservoirs with IBWT: 19.51%, N = 40; without IBWT: 15.12%, N = 31). The majority of interviewees were engaged in occupations involving direct contact with the reservoir landscape (63.41%, N = 130), such as farming and fishing, with 33.66% (N = 69) near reservoirs with IBWT and 29.76% (N = 61) near those without. Additionally, 16.59% (N = 34) were retired, and 2.44% (N = 5) of the respondents were unemployed (Table B in S1 Table).

## Riverside community perception of EGSs

In our study, we identified 29 classes of EGSs, with 78.53% falling within the provisioning section (45.09% citations from interviewees near reservoirs receiving IBWT and 33.44% from those that do not). Regulation and maintenance accounted for 7.42% of EGS citations with 3.11% from reservoirs with IBWT and 4.31% from those without, and cultural services accounted for 14.05%, with 9.42% cited by people near reservoirs with IBWT and 4.63% by those without. On average, each participant identified six EGSs (SD ± 2.61), totaling 1,237 mentions. Specifically, 707 EGSs were mentioned by individuals near reservoirs with IBWT (Mean = 6, SD ± 2.54) and 530 by those near reservoirs without IBWT (Mean = 5, SD ± 2.68).

The most frequently cited EGSs were predominantly in the provisioning section. These included surface water for drinking, surface water used for cleaning (e.g., house cleaning), cultivated terrestrial plants for nutrition (e.g., vegetables and corn), and wild aquatic animals for nutrition (primarily fish), which together accounted for 65% (N = 804) of all EGSs mentioned (Fig. 3). Of these, 38% (N = 470) were reported by the community near reservoirs with IBWT, and 27% (N = 334) by those near reservoirs without IBWT.

The least perceived EGSs, occurring only once among the responses, included provision services from wild terrestrial plants and animals used for nutrition and groundwater used for drinking, cleaning, and irrigation. Also rarely mentioned were regulation and maintenance services like wind protection, as well as cultural services related to historical heritage and sacred or religious significance (Fig 3). These services were exclusively mentioned by individuals near reservoirs without IBWT.

Additionally, provisioning services from fibers and other materials derived from wild plants for direct use or processing (e.g., wood used to make fence posts, plants for brooms) and animals reared through in situ aquaculture for nutrition were infrequently cited. Regulation and maintenance services such as dilution by freshwater ecosystems (e.g., dilution of fishery waste), weathering processes affecting soil quality (e.g., creating fertile soil for planting), and erosion control were seldom mentioned by either group of interviewees (Fig 3).

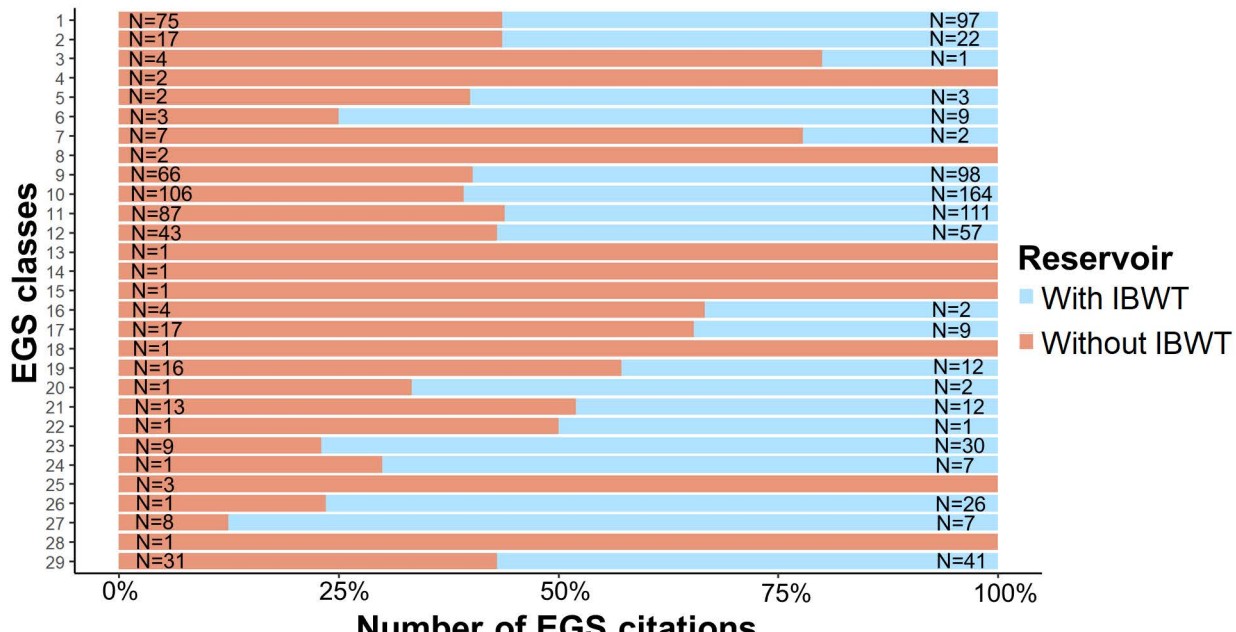

**Fig 3. EGS classes recorded in the reservoir landscapes that receive and do not receive IBWT from the São Francisco River, Paraíba, Brazil. Where: 'N' corresponds to the number of citations per EGS classes.**

## Association between sociodemographic profile, IBWT, and EGSs

Our analysis revealed that the sociodemographic factors of the interviewed populations (gender, age, education level, occupation, and monthly family income) were not significantly associated with the number of EGSs citations mentioned per person, regardless of whether they lived near reservoirs receiving IBWT or not (Table C in S1 Table). However,

significant differences were observed in the number of EGSs per section (provision, regulation and maintenance, and cultural) related to the sociodemographic variables of education level (PERMANOVA, $F_{4,340}$=4.03, $r^2$=0.02, p=0.003, Table D in S1 Table) and occupation (PERMANOVA, $F_{16,340}$=2.08, $r^2$=0.05, p=0.009, Table D in S1 Table). Specifically, individuals with elementary education identified more EGSs than those with high school education (Post-hoc, $r^2$=0.04, F=8.44, $p_{adjusted}$=0.04, Table E in S1 Table). Nonetheless, no significant differences were found among occupation classes in post-hoc tests with the Bonferroni correction (Table F in S1 Table). Furthermore, interactions with the São Francisco River water transfer did not influence the number of perceived EGSs by riverside communities (Mann-Whitney, H=4595.5, p=0.16) or the number of EGSs per section (PERMANOVA, $F_{2, 340}$=0.99, $r^2$=0.003, p=0.37).

## Association between EGSs, IBWT and LULC

The most frequently perceived EGS classes within the CICES typology were associated with the water and forest formation, each with 16 EGS classes identified by the participants. In terms of EGS counts by LULC type, the water was the most prominent (69.28%, N=857), followed by land use mosaic (16.65%, N=206). In contrast, built-up areas (3.45%, N=1), bare land (3.45%, N=1), and aquaculture (10.34%, N=3) within the reservoir landscapes were perceived as offering the fewest EGSs by the riverside communities (Fig 4, Table G in S1 Table).

IBWT from the São Francisco River did not influence the number of EGSs associated with LULC types perceived by the population (PERMANOVA, $F_{6,483}$=1.02, $r^2$=0.005, p=0.39). Nevertheless, significant differences were found in provisioning (PERMANOVA, $F_{6,483}$=168.62, $r^2$=0.68, p=0.0001), regulation and maintenance (PERMANOVA, $F_{6,483}$=34.20, $r^2$=0.30, p=0.0002), and cultural services (PERMANOVA, $F_{6,483}$=10.13, $r^2$=0.11, p=0.0002) associated with LULC (Table H in S1 Table). The post-hoc test with Bonferroni correction showed that provisioning services associated with the water LULC were mentioned more frequently than other LULC types (Post-hoc, $p_{adjusted}$=0.02). Regulation and maintenance services were most often associated with forest formation, showing a significant difference compared to water, land-use mosaics, and floodplains (Post-hoc, $p_{adjusted}$=0.02). Regarding cultural services, forest formation represented more EGS citations than land use mosaics and built-up areas (Post-hoc, $p_{adjusted}$=0.02). Built-up areas were cited for more cultural EGSs than bare land, land use mosaics, and aquaculture (Post-hoc, $p_{adjusted}$=0.02, Table I in S1 Table).

A significant difference in regulation and maintenance services was also observed between populations living near reservoirs with and without IBWT (PERMANOVA, $F_{6,483}$=8.07, $r^2$=0.01, p=0.003, Table H in S1 Table). Communities near reservoirs that did not receive IBWT cited more EGSs for regulation and maintenance services, particularly those related to temperature and humidity regulation, such as thermal comfort from tree shading, attributed to forest formation (13.19%, N=12). Additionally, maintaining nursery populations and habitats, such as biodiversity, was associated with the water (3.30%, N=3), forest formation (8.80%, N=8), and land use mosaic (1.10%, N=1) LULC types.

Furthermore, cultural services (PERMANOVA, $F_{6,483}$=7.37, $r^2$=0.01, p=0.006, Table H in S1 Table) were predominantly reported by those living near reservoirs with IBWT (67.27%, N=111). Aesthetic experiences associated with water (7.88%, N=13), forest formation (6.67%, N=11), and floodplains (1.21%, N=2) LULC types, and recreational activities related to water LULC (e.g., swimming in the reservoir and recreation) were significantly more mentioned in this group (24.85%, N=41, Table J in S1 Table).

## Association of EGSs with SDGs

In this study, 29 classes of EGSs were linked to 11 SDGs, with 25,932 number of connections (NC) identified between EGSs cited by the communities and SDGs across three SDG domains. The domain for securing the basic material needs of human beings was associated with EGSs through SDG 1- No Poverty (19 EGS classes), SDG 3- Good Health and Well-being (21 EGS classes), and SDG 8- Decent Work and Economic Growth (20 EGS classes).

The domain focused on pursuing common human well-being was represented by SDG 2- Zero Hunger (24 EGS classes), SDG 6- Clean Water and Sanitation (12 EGS classes), SDG 7 Affordable and Clean Energy (1 EGS class), and

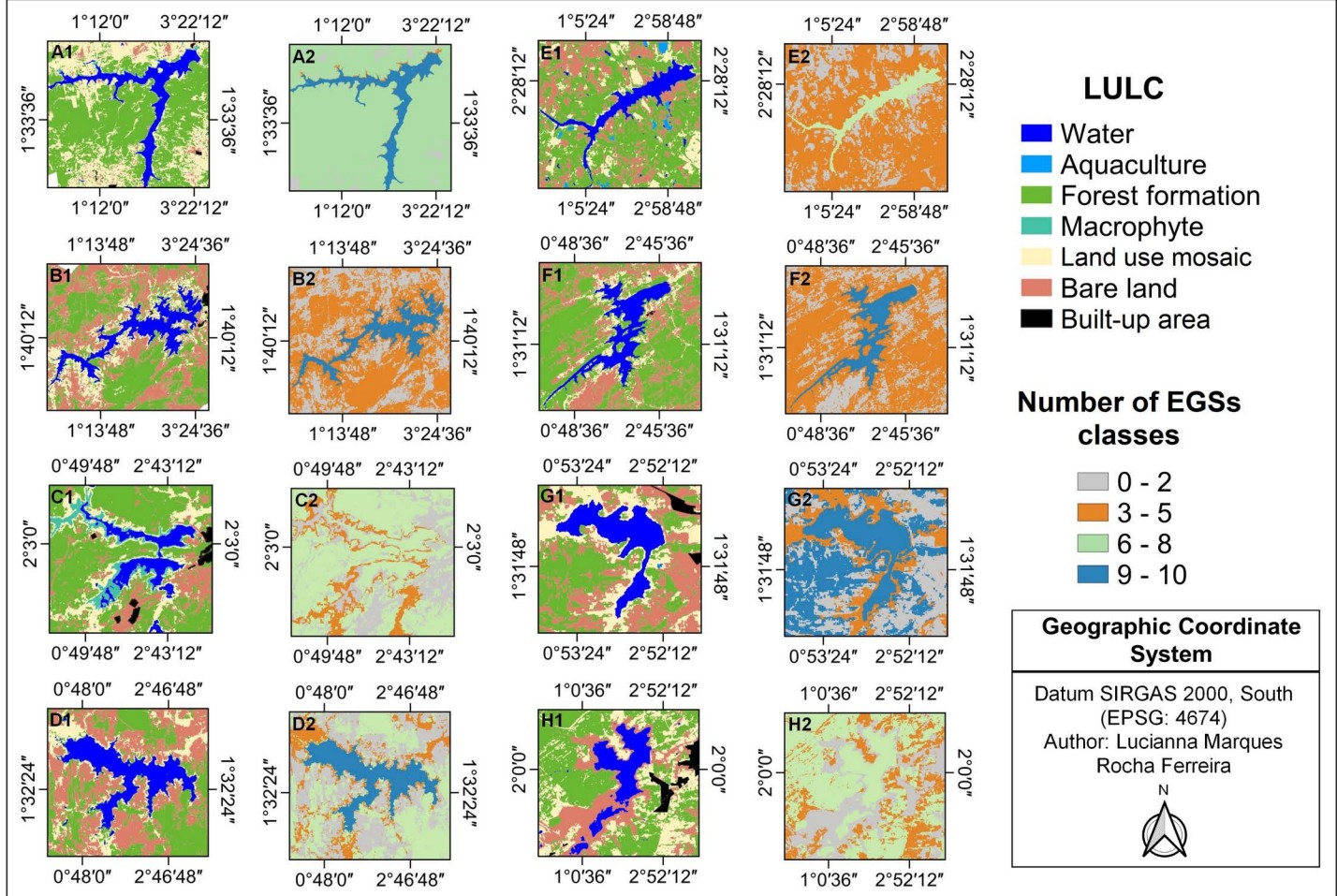

**Fig 4. Thematic map illustrating LULC alongside the number of EGSs classes for reservoirs receiving IBWT (A = Argemiro de Figueiredo; B = Epitácio Pessoa; C = Camalaú; and D = Poções) and those that do not (E = Araçagi; F = Cordeiro; G = Sumé; and H = Taperoá) in Brazil. Author: Lucianna Marques Rocha Ferreira.** The maps labeled A1, B1, C1, D1, E1, F1, G1, and H1 depict the LULC types, while the maps labeled A2, B2, C2, D2, E2, F2, G2, and H2 represent the corresponding number of EGSs classes identified in each reservoir landscape.

SDG 15- Life on Land (29 EGS classes). Furthermore, the domain aimed at coordinating sustainable governance policies encompassed SDG 9- Industry, Innovation and Infrastructure (9 EGS classes), SDG 11- Sustainable Cities and Communities (19 EGS classes), SDG 12- Responsible Consumption and Production (19 EGS classes), and SDG 13- Climate Action (29 EGS classes).

SDG 1 (13.20%, NC = 3423), SDG 2 (10.91%, NC = 2828), SDG 3 (11.68%, NC = 3028), SDG 12 (12.24%, NC = 3173), and SDG 15 (23.70%, NC = 6147) accounted for 71.73% of the community-perceived linkages between EGSs and SDGs (Table K in S1 Table). Provisioning services, accounting for 85.52% (NC = 22176) of perceived ecosystem benefits, were most frequently mentioned and strongly associated with SDGs (PERMANOVA, $F_{17,2913}$ = 36.39, $r^2$ = 0.08, p = 0.0001, Table 1, Fig. 5). These have greater potential to contribute to the achievement of the SDGs, particularly SDG 15, 12, 1, 3, and 2 (Fig 5, Table K in S1 Table), in ascending order of the number of associations between EGSs perceived by the surveyed population and the SDGs, regardless of whether the reservoirs receive IBWT or not (PERMANOVA, $F_{17,2913}$ = 0.41, $r^2$ = 0.0009, p = 0.9, Table 1, Fig 5).

Table 1. Results of the PERMANOVA statistical test for interaction between IBWT variables, number of EGSs per section, and SDGs.

| Interaction | Df[a] | R²[b] | F[c] | p-value |
|---|---|---|---|---|
| IBTW variable, and number of EGSs | 1 | 0.001 | 7.78 | 0.005 |
| EGSs section, and number of EGSs | 2 | 0.16 | 608.68 | 0.0001 |
| SDGs and number of EGSs | 10 | 0.39 | 307.10 | 0.0001 |
| IBTW variable, EGSs section, and number of EGSs | 2 | 0.001 | 4.11 | 0.02 |
| IBTW variable, SDGs, and number of EGSs | 10 | 0.0006 | 0.46 | 0.91 |
| EGSs section, SDGs, and number of EGSs | 17 | 0.08 | 36.39 | 0.0001 |
| IBTW variable, EGSs section, SDGs, and number of EGSs | 17 | 0.0009 | 0.41 | 0.9 |
| Residual | 2854 | 0.37 | | |
| Total | 2913 | 1 | | |

[a]Df = degrees of freedom.

[b]R² = coefficients of determination.

[c]F = F value by permutation.

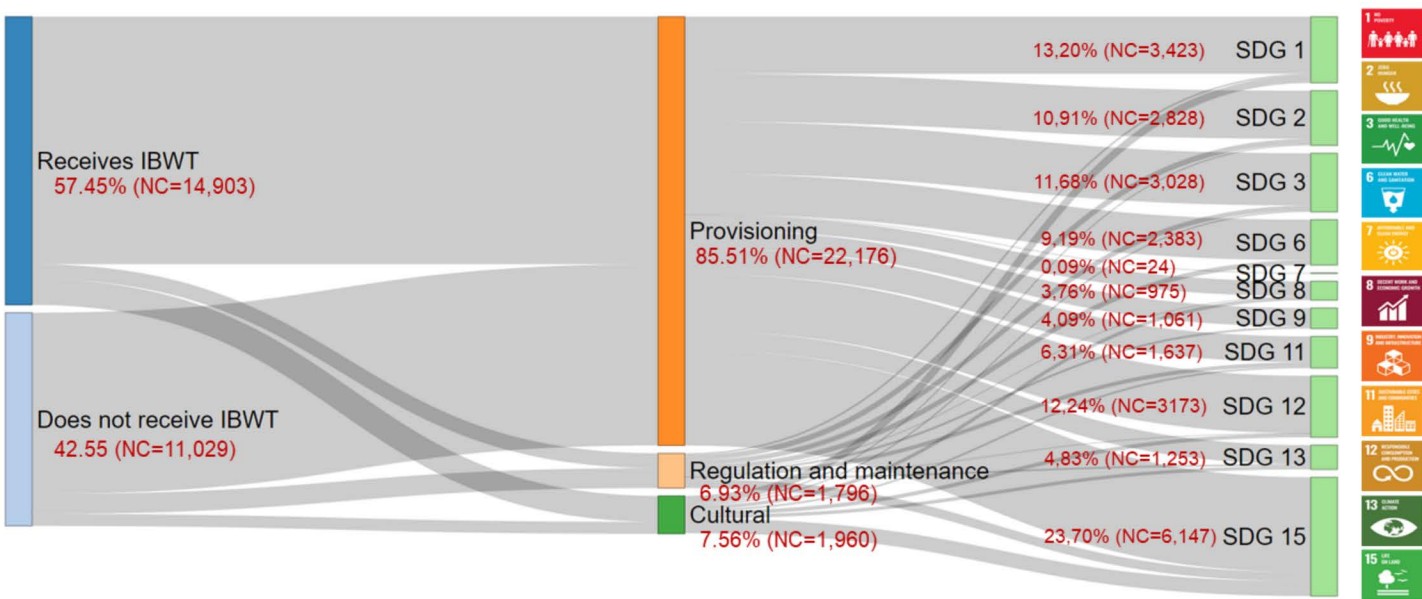

Fig 5. Links between the number of EGSs recognized by respondents residing close to the reservoirs that do and do not receive IBWT, categorized by EGS sections (provisioning, regulation and maintenance, and cultural), and their relationship with SDGs. Where: 'NC' corresponds to the number of connections between EGSs and SDGs.

## Discussion

### Influence of the IBTW variable and sociodemographic factors on riverside community perceptions of EGSs

The presence of IBWT from the São Francisco River and sociodemographic factors did not lead to significant differences in EGS perception among the riverside communities. This observation does not support our first hypothesis. To understand this, it is essential to consider the recent challenges faced by these communities. The communities across all studied reservoirs suffered the impacts of an extended drought from 2012 to 2017, marked as the most severe in over five decades, causing some regional reservoirs to reach water collapse [22,23,41]. The recovery of reservoir volumes and the

resumption of agricultural activities, previously halted during the extreme drought, began gradually following the implementation of the IBWT from the São Francisco River in 2017 [21] and subsequent rainfall in 2018 [22]. However, these developments are inadequate to restore water quality and aquatic ecological dynamics [21,41], limiting the opportunity for the riverside community to engage with and adapt to the IBWT-enhanced reservoir landscapes. Consequently, these communities had insufficient time to cultivate personal and cultural experiences that could enhance the recognition of additional EGSs.

The perceived benefits of the São Francisco River IBWT did not result in distinctive individual experiences that could allow the identification of a greater number of EGSs and EGS classes per section among participants living near reservoirs with or without IBWT. Therefore, our second hypothesis was not supported. This result was evidenced by the similarity in the number of EGS classes identified by the interviewees.

Provisioning services were the most frequently cited across all studied reservoir landscapes. These services are more immediately perceptible and essential for survival, making them easier for individuals to identify and remember [6,42]. Our study showed that riverside communities discern EGS based on their routine needs and ongoing interactions with the surrounding environment, corroborating findings from some semiarid regions in Spain [42] and West African reservoir landscapes [6]. Thus, local engagement with the reservoir landscapes were similar regardless of the IBWT.

The education levels of the interviewees provided different experiences with the reservoir landscape, regardless of the presence of IBWT. Significant differences were observed in the number of the EGS citations per section (provision, regulation and maintenance, and cultural) depending on the level of education of the interviewees, partially affirming our first hypothesis. The main difference was observed between people with primary versus secondary education, with the former perceiving more EGSs than the latter. The research developed by [6] aimed to understand ecosystem service and disservice perceptions among farmers in four reservoir landscapes in West Africa. They found that less formally educated, younger people valued EGSs provided by water resources and agricultural activities more than their more educated, older counterparts, due to a more direct reliance on ecosystem functions and available natural resources [6].

People with lower levels of education tend to have occupations closely linked to the countryside (e.g., farming, livestock and fishing) and hence have more frequent contact with landscape elements than individuals with higher formal education levels. Interviewees predominantly cited EGSs related to basic needs - drinking water, food from cultivation and fishing - highlighting the critical role of the reservoir landscape in providing essential resources, especially in semiarid regions. Reservoirs are crucial for balancing social and economic demands [1].

The findings of this study underscore the importance of introducing educational initiatives to foster both formal and informal knowledge of riverside communities regarding biotic and abiotic elements and interactions within the local landscape. Enhancing the perceptions of communities regarding EGSs is crucial [43]. When combined with policy interventions, this approach is vital for protecting the landscape elements that are pivotal to sustainable development.

## IBWT and the association between perceived EGSs by riverside populations and LULC

The significant number of EGS classes associated with water and forest formation LULC, which confirms our third hypothesis, can be attributed to the historical scarcity of water in the study area and the crucial role of reservoirs in survival and economic activities. The highest number of EGS citations was linked to water and land use mosaics (agriculture and livestock), in contrast to [6], who reported a greater diversity of EGSs in native forests and land-use mosaics. They posited that individual involved in agriculture perceive more EGSs from cultivated land because they rely on these practices and the local environment [6]. We infer that EGSs related to water, forest formation, and fertile floodplain soils for agriculture underpin the livelihoods of the interviewed riverside communities.

Distinctive perceptions of regulation and maintenance and cultural services associated with LULC emerged in this research. Cultural services that promote well-being, stress relief, and recreational activities are recognized most by communities near reservoirs with IBWT. The role of IBWT in sustaining adequate water levels supports landscape elements

conducive to recreational and contemplative activities. Communities near reservoirs with IBWT have enhanced access to water-associated EGSs, which, when basic human needs are met, allow experiences that promote social relationships, self-awareness, and spiritual well-being. Communities living near reservoirs without IBWT experience more intense seasonal fluctuations and perceive a greater number of regulation and maintenance services. The study elaborated by [44] observed that diminished engagement with rural landscapes and traditional environmental practices can reduce the social perception of cultural services. Therefore, the relevance of regulation and maintenance services - specifically, maintaining populations and nursery habitats associated with water, forest formation, and land use mosaics - was more noticeable among communities without IBWT. This reflects their adaptation to climatic drought cycles typical of the semiarid region of Northeastern Brazil [23] and the lack of regular water volume control, unlike in landscapes with IBWT. In some of the reservoir landscapes studied, [16] demonstrated that riverside communities adapt to different forms of water access and usage purposes according to seasonality and variations in water availability and quality. This indicates that the need for ecosystem regulation and maintenance is perceived more acutely than cultural services in these areas. This group also identified that regulation and maintenance services are critical for preventing desertification and climate change impacts, echoing the findings of [45] in Portugal.

## EGSs that can support strategies for achieving SDGs

Securing basic material needs, a critical aspect of human survival, remains central to the implementation of the SDGs established by the United Nations [1]. The reservoir landscape plays a vital role in this respect, aligning with the achievement of SDGs such as No poverty (SDG 1), Zero Hunger (SDG 2), Good Health and Well-being (SDG 3), Clean Water and Sanitation (SDG 6), Affordable and Clean Energy (SDG 7), Decent Work and Economic Growth (SDG 8), Industry, Innovation, and Infrastructure (SDG 9), Sustainable Cities and Communities (SDG 11), Responsible Consumption and Production (SDG 12), Climate Action (SDG 13) and Life on Land (SDG 15). This linkage is supported by the perceptions of EGSs expressed by the interview participants in this study and their associations between EGSs and the SDGs. Therefore, these goals have the potential to sustainably address the challenges posed by water scarcity in semiarid regions, which not only affects the availability of water for human supply but also poses socioeconomic challenges, as underscored by [1] in their review of reservoir ecosystems and SDGs, thereby fulfilling their social role in supplying water for human needs.

Our study revealed that perceptions of riverside communities were associated with EGSs focused on drinking water and personal hygiene, wild aquatic animals used for nutritional purposes, and cultivated terrestrial plants grown for food. Provisioning services were linked to the targets of SDG 1, 2, 3, 12, and 15. Integrating the EGSs of the reservoir landscape into the SDGs framework will provide basic material security for water and food, contributing to the sustainable development of semiarid regions. This approach aligns with [3], who assessed the knowledge of experts on the link between SDGs and EGSs in various macro-regions (Asia, Europe, North America, Oceania, Latin America, the Caribbean, and Africa) and found that the most prioritized EGSs were related to people's basic material needs, such as SDG 1, 2, and 6.

Conversely, the exploitation of provisioning services by riverside communities is linked to their use. [46] showed that effluent discharge, deforestation, and overfishing are the main threats highlighted by local communities in four reservoirs in the semiarid region of Brazil, which are responsible for environmental degradation and compromised reservoir conservation.

Based on the results of our research and the literature [21,30,47], we believe that in order to achieve the goals of the 2030 Agenda, the landscape management of reservoirs—both those that receive and those that do not receive IBWT—should be based on (i) participatory monitoring, where the local population actively participates in data collection and environmental monitoring, integrating knowledge from local experience with scientific expertise; and (ii) adaptive management of the region's climate variability, particularly in semi-arid areas. However, this management must consider the specific characteristics of the reservoir landscapes that receive and do not receive IBWT.

It is evident that IBWT projects directly impact access to and rights over water usage for local populations, particularly during drought periods. These projects increase the number of users accessing the resource and the associated ecosystem services (EGS). In this sense, reservoirs receiving water from IBWT projects benefit from water infrastructure that ensures continuous access to water, thereby fostering local and regional economic development by enabling a greater diversity of socio-economic activities, especially agriculture, fishing, and recreation, as evidenced by this study. Based on our findings, we believe that landscape management of reservoirs, to achieve the goals of the 2030 Agenda, should consider the following: (i) integrating the local community into governance processes; (ii) preventing system overload due to increased users and socio-economic activities, particularly during droughts; (iii) managing water use conflicts to reduce inequalities in access and water rights; (iv) creating policies to expand local ecotourism, generating income while promoting culture, environmental conservation, and EGSs; (v) developing and strengthening environmental education activities.

In contrast, populations living near reservoirs that do not benefit from IBWT projects have more restricted access and rights to water, making them more vulnerable to water scarcity and the effects of drought. Therefore, management in these areas should consider: (i) integrating the local community into the governance process; (ii) stricter adherence to system limits, taking into account the supply and demand for EGSs, especially provisioning and regulation and maintenance services; (iii) reducing inequalities in water access through the strengthening of practices that optimize and store water (e.g., cisterns); (iv) developing income-generation policies, respecting climatic seasonality; (v) strengthening government social assistance programs during drought periods; and (vi) developing and strengthening environmental education activities.

## Conclusion

The perceptions of EGSs by individuals were comparable in reservoir landscapes with and without IBWT, particularly regarding the number and diversity of EGS citations. Even without IBWT, riverside communities depend on certain fundamental EGSs for survival. However, prioritizing investment in both formal and informal education within riverside communities is vital for fostering knowledge that helps people better recognize landscape elements, thereby increasing the number of perceived EGSs per person and promoting environmental stewardship. Different education levels were evident when analyzing the EGSs section (provision, regulation and maintenance, and cultural), correlating with individual dependence on the structure and functioning of the ecosystems, especially for provisioning services such as water and food.

The coexistence of riverside communities with the IBWT project influenced the perceptions of individuals of landscape elements, affecting their social perceptions of EGS sections. Residents near IBWT reservoirs identified more cultural services, while those near non-IBWT reservoirs recognized more regulation and maintenance services. The that receives IBWT reservoirs provided greater access to and rights over water use for the surrounding communities, offering these riverside populations new daily experiences with water. The increased water access enabled recreational and tourism activities (cultural services), which are limited in reservoirs that do not receive IBWT.

Water and forest formation, the LULC types with the highest number of EGSs citations, reveal the social importance of ensuring basic needs and well-being, thus serving as critical of the population. Therefore, these are key areas for policy coordination and sustainable governance in line with the 2030 Agenda. Perceptions of reservoir landscape elements and benefits relate to fundamental human experiences and necessities, particularly provisioning services.

This study showed that provisioning benefits, when well-managed, have significant potential to contribute to the SDGs focused on securing basic needs. As such, it is prudent to manage and monitor reservoir landscapes and river basins involved in IBWT projects to identify, map, and evaluate EGSs with synergies to the SDGs. Determining which EGSs and associated LULC prioritize conservation will lead to the achievement of multiple SDGs.

## Supporting information

**S1 Table. Database and complementary results.**
(XLSX)

## Acknowledgments

The authors thank all members of the Laboratório de Ecologia de Bentos (Universidade Estadual da Paraíba) who participated in the community interview phase. We also express our gratitude to the interviewees for their willingness and kindness to participate in this research. Additionally, we sincerely thank the anonymous reviewers for their thoughtful and constructive suggestions, which greatly improved this work.

## Author contributions

**Conceptualization:** Lucianna Marques Rocha Ferreira, Franciely Ferreira Paiva, Joseline Molozzi.

**Formal analysis:** Lucianna Marques Rocha Ferreira, Franciely Ferreira Paiva, Maria Eduarda Santana Veríssimo.

**Funding acquisition:** José Etham de Lucena Barbosa, Joseline Molozzi.

**Investigation:** Lucianna Marques Rocha Ferreira, Franciely Ferreira Paiva, Maria Eduarda Santana Veríssimo.

**Methodology:** Lucianna Marques Rocha Ferreira, Joseline Molozzi.

**Project administration:** Lucianna Marques Rocha Ferreira.

**Supervision:** José Etham de Lucena Barbosa, Joseline Molozzi.

**Visualization:** Lucianna Marques Rocha Ferreira, Franciely Ferreira Paiva, Maria Eduarda Santana Veríssimo, Lívia Maria Osório de Sousa, Evaldo de Lira Azevêdo.

**Writing – original draft:** Lucianna Marques Rocha Ferreira.

**Writing – review & editing:** Lucianna Marques Rocha Ferreira, Franciely Ferreira Paiva, Maria Eduarda Santana Veríssimo, Lívia Maria Osório de Sousa, Evaldo de Lira Azevêdo, José Etham de Lucena Barbosa, Joseline Molozzi.

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
