## [Decision Letter · Decision Letter 0]

29 Dec 2024

PONE-D-24-27908Community perception differences regarding of ecosystem goods and services in landscape reservoirs with and without Inter-Basin Water Transfer: implications for Sustainable Development GoalsPLOS ONE

Dear Dr. Ferreira,

Thank you for submitting your manuscript to PLOS ONE. After careful consideration, we feel that it has merit but does not fully meet PLOS ONE’s publication criteria as it currently stands. Therefore, we invite you to submit a revised version of the manuscript that addresses the points raised during the review process.

We look forward to receiving your revised manuscript.

Kind regards,

Mattias Gaglio, PhD

Academic Editor

PLOS ONE

Journal Requirements:

4. We note that Figures 1,2 and 4 in your submission contain map/satellite images which may be copyrighted. All PLOS content is published under the Creative Commons Attribution License (CC BY 4.0), which means that the manuscript, images, and Supporting Information files will be freely available online, and any third party is permitted to access, download, copy, distribute, and use these materials in any way, even commercially, with proper attribution. For these reasons, we cannot publish previously copyrighted maps or satellite images created using proprietary data, such as Google software (Google Maps, Street View, and Earth). For more information, see our copyright guidelines: http://journals.plos.org/plosone/s/licenses-and-copyright.

 a. You may seek permission from the original copyright holder of Figures 1,2 and 4  to publish the content specifically under the CC BY 4.0 license. 

Reviewers' comments:

Reviewer's Responses to Questions

**Comments to the Author**

1. Is the manuscript technically sound, and do the data support the conclusions?

Reviewer #1: Yes

Reviewer #2: Yes

2. Has the statistical analysis been performed appropriately and rigorously? 

Reviewer #1: No

Reviewer #2: Yes

3. Have the authors made all data underlying the findings in their manuscript fully available?

Reviewer #1: Yes

Reviewer #2: Yes

4. Is the manuscript presented in an intelligible fashion and written in standard English?

Reviewer #1: Yes

Reviewer #2: Yes

5. Review Comments to the Author

Reviewer #1: Comments on the paper entailed on the “Community perception differences regarding of ecosystem goods and services in landscape reservoirs with and without Inter-Basin Water Transfer: implications for Sustainable Development Goals” submitted to the “PLOS ONE”. The paper mainly focuses on the perception on ecosystem services from landscape reservoirs.

I have some basic issues that need to be resolved. My major concerns are as follows:

• Abstract section: In the abstract section, it lacks empirical findings from the results. The findings need to be supported with empirical analysis and findings.

• Introduction: In the introduction, I missed the linkages between ES goods and services and SDGs in the introduction section. How the SDGs can be integrated with the ES incorporated in this study. It needs to be elaborated in the introduction section. Line from 75 to 83 is very short and it needs to be extended to establish the linkages between ES and SDGs in the introduction section. I missed the novelty of the study. It needs to be incorporated at the end of the introduction section.

• Materials and methods: The methods sections are very hard to understand. I have few issues to be clarified in the method sections. Firstly, in case of the sample size determination, how the sample size was determined? It needs to be clarified. How the semi structure interview was carried out? It needs to be described in details. Where and how the SDGs have been incorporated in the method sections? I did not understand. It needs to be clarified in the method sections. Secondly, what are the different type of ES and SDG used in the study? These are not clearly mentioned in the methods section. A methodological framework needs to be developed for better understanding the research work.

• Results: Few issues need to be fixed in the results sections. Firstly, how the socio-economic factors affect to the perception of ES. What are the different factors associated to the ES perception? For me, I think the difference between river side’s (such as difference on the banks) needs to be assessed for better understanding the spatial variability. The associations of SDGs are not clear not me. It needs to be discussed.

• In the discussion section, the how policies can be helpful to achieve SDG through ES- It needs to be elaborated.

Reviewer #2: PLOS ONE

Title: Community perception differences regarding of ecosystem goods and services in

landscape reservoirs with and without Inter-Basin Water Transfer: implications for

Sustainable Development Goals

Introduction

This is well written but the authors should explain what they mean by Inter-Basin Water Transfer as it is an important variable in the study.

Water is used as an entry point to the writing of this manuscript and so it would be important if the status and distribution of water in the areas studied is explained. This is only done briefly by authors explaining in the discussion line 472 to 475.

Materials and methods

The authors should consider explaining the role of LULC in the study, is this introduced as a factor impacting the reservoirs? i.e. in line 87, 88, 89 and 90 the author (s) introduces the LULC and states that “The perceptions of riparian communities highlight the essential landscape and ecosystem benefits that are crucial for sustaining life, including water and nutrients, cultural and educational values, and personal environmental experiences”. It is not clear if changes in landscapes impacts water reservoirs which then impact the provisioning of EGS. The authors should consider clarifying

Also the author should explain if the study mapped EGS using landscape based approaches and reservoirs was one of the LULC. In short what did the author(s) consider as LULC.

There is also the interchangeable use of water bodies, reservoirs, aquatic environment . I would advice that the authors stick to this reservoir landscape elements to avoid confusing the readers as there are many variables and treatments in the study and the more different concepts or terms are used interchangeably, the more easier it is to loose a reader. The use of the reservoir landscape elements will also make it easier for the authors to explain the reason for introducing the LULC in the study.

Are interviewees the same as riverside communities in line 566

Discussion and Conclusions

These are well written but the sections could be tightened by the authors explaining confounding factor like do reservoirs that receive IBWT and those that are non IBWT reservoirs impact access and water use rights of the local communities and what does this mean for the management of the reservoirs.

General

The manuscript is well written and interesting study.

6. PLOS authors have the option to publish the peer review history of their article (what does this mean? ). If published, this will include your full peer review and any attached files.

**Do you want your identity to be public for this peer review?** For information about this choice, including consent withdrawal, please see our Privacy Policy .

Reviewer #1: No

Reviewer #2: No

---

## [Author Response · Author response to Decision Letter 1]

3 Feb 2025

Please accept the revised version of our manuscript entitled “Community perception differences regarding ecosystem goods and services in reservoir landscapes with and without Inter-Basin Water Transfer: implications for Sustainable Development Goals” for consideration for publication in PLOS ONE. We are grateful to the editor for handling our paper and to the reviewers for their important comments, suggestions, and criticisms, which were very helpful in improving our manuscript. We have carefully addressed the reviewer's comments and suggestions, and we hope the manuscript is now ready for acceptance.

Our response letter delineates point-by-point responses to all reviewers’ comments and suggestions. To facilitate the revision process, we also provide another file with marked changes (please see “revised manuscript with track changes).

We hope we have provided satisfactorily responses and improved the quality of the manuscript for its publication in PLOS ONE. We are available to perform any further changes if necessary.

- -

Reviewers' comments:

Answer: Thank you for your evaluation. We are grateful for the opportunity to improve our manuscript. We have carefully considered all the comments from the editor and reviewers, particularly to ensure that the analyses were conducted and presented with precision and rigor.

Reviewer #1

Comments on the paper entailed on the “Community perception differences regarding of ecosystem goods and services in landscape reservoirs with and without Inter-Basin Water Transfer: implications for Sustainable Development Goals” submitted to the “PLOS ONE”. The paper mainly focuses on the perception on ecosystem services from landscape reservoirs.

I have some basic issues that need to be resolved. My major concerns are as follows:

• Abstract section: In the abstract section, it lacks empirical findings from the results. The findings need to be supported with empirical analysis and findings.

Answer: Thank you for your recommendation. We have incorporated the suggested information into the Abstract. You can find these updates on lines 20 to 32.

• Introduction: In the introduction, I missed the linkages between ES goods and services and SDGs in the introduction section. How the SDGs can be integrated with the ES incorporated in this study. It needs to be elaborated in the introduction section. Line from 75 to 83 is very short and it needs to be extended to establish the linkages between ES and SDGs in the introduction section. I missed the novelty of the study. It needs to be incorporated at the end of the introduction section.

Answer: Thank you for your observation. We have restructured the introduction to better clarify the connection between Ecosystem Goods and Services (EGSs) and the Sustainable Development Goals (SDGs), as well as the changes in Land Use and Land Cover (LULC) resulting from Inter Basin Water Transfer (IBWT). These revisions can be found on lines 61 to 67 and 83 to 97. Additionally, we have highlighted the novelty of our study at the end of the introduction section (line 107 and lines 119 to 125).

• Materials and methods: The methods sections are very hard to understand. I have few issues to be clarified in the method sections.

Firstly, in case of the sample size determination, how the sample size was determined? It needs to be clarified.

Answer: Thank you for the opportunity to clarify this aspect of our methodology. The sample size for the interviews was determined using a 200 meters territorial boundary around each reservoir, following the approach described by Azevêdo et al. (2022). We visited all households within this boundary and interviewed all individuals aged 18 or older who agreed to participate. This clarification has been added to lines 204 and 205. Additionally, we have provided more detailed information about the research steps, as indicated in lines 193 to 198. We also revised Figure 2, "Scheme for obtaining and processing primary data," to make the methodology clearer and easier to understand (see line 209). Furthermore, we have renamed the section "Sociodemographic data" to "Semi-structured form: sociodemographic data" to make it more intuitive and aligned with Figure 2 (see line 224).

Reference

Azevêdo E de L, Alves RRN, Dias TLP, Álvaro ÉLF, Barbosa JE de L, Molozzi J. Perception of the local community: What is their relationship with environmental quality indicators of reservoirs? Cañedo-Argüelles Iglesias M, organizador. PLoS One. 2022;17: e0261945. doi:10.1371/journal.pone.0261945

How the semi structure interview was carried out? It needs to be described in details.

Answer: Thank you for highlighting this aspect of the methodology. We have now provided a more detailed description of how the semi-structured interviews were conducted. Please refer to lines 225 to 229 for these updates. Additionally, we considered providing more details on participatory mapping, as noted in lines 247 to 249.

Where and how the SDGs have been incorporated in the method sections? I did not understand. It needs to be clarified in the method sections.

Answer: We appreciate the reviewer’s comment, and as suggested, we have clarified this information in the text. These clarifications have been incorporated as detailed below: The SDGs were incorporated into the section "Participatory Mapping: Identification of EGSs and their association with SDGs" in lines 257 to 259. We clarified this information in the text by renaming the section from "Participatory Mapping: identification of EGSs and SDGs" to "Participatory Mapping: identification of EGSs and their association with SDGs" in line 236 to 237.

Secondly, what are the different type of ES and SDG used in the study? These are not clearly mentioned in the methods section. A methodological framework needs to be developed for better understanding the research work.

Answer: Thank you for your comment. The types of EGSs used in the study were based on the Common International Classification of Ecosystem Services (CICES), version 5.1, as mentioned in lines 250 to 252 of the manuscript and detailed in Table A of the supplementary material (S1). The SDGs were based on the 17 SDGs and their 169 targets, established by the United Nations in 2015. We have also improved the manuscript text to make this information clearer for the reader, as reflected in lines 257 to 259.

• Results: Few issues need to be fixed in the results sections. Firstly, how the socio-economic factors affect to the perception of ES. What are the different factors associated to the ES perception? For me, I think the difference between river side’s (such as difference on the banks) needs to be assessed for better understanding the spatial variability. The associations of SDGs are not clear not me. It needs to be discussed.

Answer: Thank you for your feedback. According to the findings of our research, only the socioeconomic factor "education level" significantly influences the perception of the riverside community regarding the number of EGSs mentioned, regardless of whether the person lives near a reservoir receiving IBWT or not.

Upon reviewing the "Association between sociodemographic profile, IBWT, and EGSs" section, we recognized that it was not clear that our study considered two groups: those living near reservoirs that receive IBWT and those living near reservoirs that do not. Therefore, we have revised the text to ensure this distinction is clear to the reader (see line 421).

Regarding the SDGs, our hypothesis iv was that, “among the perceived services, provisioning services are the most recognized by populations, playing a crucial role in achieving the SDGs, regardless of their proximity to reservoirs with or without IBWT”. Therefore, our study did not focus on directly associating sociodemographic factors with the SDGs. Instead, we examined the relationship between EGSs and the SDGs, following the domains suggested by Wood et al. (2018).

However, we agree with the reviewer’s suggestions and have made the necessary clarifications. We have clarified the connection between EGSs and the SDGs in the introduction (lines 61 to 67 and 83 to 97), the materials and methods sections (figure 2 lines 209 to 211 and 250 to 270), in the results (see lines504 to 508), and have reinforced this aspect in the discussion (see lines 614 to 622). We believe these clarifications provide a solid foundation for readers to fully understand this relationship in our discussion.

Reference

Wood SLR, Jones SK, Johnson JA, Brauman KA, Chaplin-Kramer R, Fremier A, et al. Distilling the role of ecosystem services in the Sustainable Development Goals. Ecosyst Serv. 2018;29: 70–82. doi:10.1016/j.ecoser.2017.10.010

• In the discussion section, the how policies can be helpful to achieve SDG through ES- It needs to be elaborated.

Answer: Thank you for your valuable recommendation. We completely agree with your perspective and have expanded on this point in the discussion to further elaborate on how policies can help achieve the SDGs through Ecosystem Services (see lines 628 to 629 and 642 to 673). Additionally, we have integrated feedback from Reviewers 1 and 2 into this paragraph, which we believe has added significant depth and strength to our discussion. We hope these revisions address your concerns.

Reviewer #2: PLOS ONE

Title: Community perception differences regarding of ecosystem goods and services in landscape reservoirs with and without Inter-Basin Water Transfer: implications for Sustainable Development Goals

Introduction

This is well written but the authors should explain what they mean by Inter-Basin Water Transfer as it is an important variable in the study. Water is used as an entry point to the writing of this manuscript and so it would be important if the status and distribution of water in the areas studied is explained. This is only done briefly by authors explaining in the discussion line 472 to 475.

Answer: Thank you for your observation. We included the definition of IBWT and expanded on this topic to clarify the connection between the benefits and drawbacks of IBWT, landscape changes, ecosystem services, and the SDGs (see lines 73 to 97). Additionally, we have included more details on water distribution in the studied areas within the materials and methods section (see lines 156 to 171). We hope this provides a clearer understanding of the role of water in the context of the study.

Materials and methods

The authors should consider explaining the role of LULC in the study, is this introduced as a factor impacting the reservoirs? i.e. in line 87, 88, 89 and 90 the author (s) introduces the LULC and states that “The perceptions of riparian communities highlight the essential landscape and ecosystem benefits that are crucial for sustaining life, including water and nutrients, cultural and educational values, and personal environmental experiences”. It is not clear if changes in landscapes impacts water reservoirs which then impact the provisioning of EGS. The authors should consider clarifying.

Answer: Thank you for the opportunity to clarify this point. While this study did not specifically aim to evaluate changes in LULC and their direct impact on reservoirs and the provision of EGSs, we recognize—based on existing literature—that LULC changes resulting from the implementation of IBWT projects do affect EGS provision. We have made revisions to the introduction, particularly in lines 73 to 97, to better address this context and clarify the connection. Additionally, we have clarified this point in the Materials and Methods section, on lines (193 to 198). We hope these revisions address the reviewer’s concerns.

Also the author should explain if the study mapped EGS using landscape based approaches and reservoirs was one of the LULC. In short what did the author(s) consider as LULC.

Answer: Thank you for the opportunity to clarify this. We mapped the EGSs based on responses from participatory mapping conducted with the riverside population living within 200 meters of the reservoir, using the reservoir landscape approach. This information was clarified in lines 196 to 198 and line 274. It is important to note that reservoirs, watercourses, and other surface water bodies were classified under the LULC water category, as explained in line 298. We have carefully reviewed these points throughout the manuscript (see lines17, 278, 288, 310, 313, 322, and 352).

Additionally, we have renamed the section “Land use and land cover classification” to “Land use and land cover classification: mapping of EGSs and their association with LULC” to make it clearer and more consistent with Figure 2 (see lines 272 to 273).

There is also the interchangeable use of water bodies, reservoirs, aquatic environment . I would advice that the authors stick to this reservoir landscape elements to avoid confusing the readers as there are many variables and treatments in the study and the more different concepts or terms are used interchangeably, the more easier it is to loose a reader. The use of the reservoir landscape elements will also make it easier for the authors to explain the reason for introducing the LULC in the study.

Answer: Thank you for your valuable comment. We appreciate your attention to this detail. We agree that the use of multiple terms can cause confusion in the text. Therefore, we clarified this point in line 298 of the manuscript and revised the entire text, removing the terms "aquatic environment" and "water body" (lines 299, 437, 438, 461, 581, 585, and 694) to focus on the LULC classification of the reservoir landscapes (“water”), making the text clearer and more accessible for the reader. It is worth mentioning that the reservoir landscapes studied are located in a semi-arid region where it is common for people to construct small or medium-sized dams to store surface water on their properties. These water bodies are part of the reservoir landscapes analyzed.

Are interviewees the same as riverside communities in line 566.

Answer: Thank you for your helpful observation. Yes, they are the same. To avoid confusion, we have changed "communities" to "interview participants" in the text and we have restructured the paragraph in the discussion section to make the information clearer for the reader (see lines 621 to 622). Thank you for your observation.

Discussion and Conclusions

These are well written but the sections could be tightened by the authors explaining confounding factor like do reservoirs that receive IBWT and those that are non IBWT reservoirs impact access and water use rights of the local communities and what this mean for the management of the reservoirs.

Answer: Thank you for your thoughtful suggestion. We completely agree with your perspective and have addressed this point in the manuscript. We expanded our discussion to explain how reservoirs receiving Inter-Basin Water Transfer (IBWT) and those that are not may affect local communities' access to and use of water, as well as the implications for reservoir management (see lines 642 to 673). Additionally, this perspective was incorporated into the conclusion (see lines 690 to 693).

Furthermore, upon reviewing the topic "Influence of LULC on riverside population perceptions of EGSs and the IBWT variable" in the discussion section, we realized that the title was not fully align with our objective and might cause confusion. To improve clarity, we have revised it to "IBWT and the association between perceived EGSs by riverside populations and LULC" (see lines 576 and 577). We hope these changes address your concerns and improve the manuscript's clarity.

General

The manuscript is well written and interesting study.

Answer: Thank you for your dedication in reviewing our work review and positive opinion on our study. We also appreciate your recommendations; they helped improve the quality of our manuscript.

---

## [Decision Letter · Decision Letter 1]

13 Apr 2025

Community Perception Differences Regarding Ecosystem Goods and Services in Reservoir Landscapes With and Without Inter-Basin Water Transfer: Implications for the Sustainable Development Goals

PONE-D-24-27908R1

Dear Dr. Ferreira,

We’re pleased to inform you that your manuscript has been judged scientifically suitable for publication and will be formally accepted for publication once it meets all outstanding technical requirements.

Kind regards,

Mattias Gaglio, PhD

Academic Editor

PLOS ONE

Additional Editor Comments (optional):

Reviewers' comments:

Reviewer's Responses to Questions

**Comments to the Author**

1. If the authors have adequately addressed your comments raised in a previous round of review and you feel that this manuscript is now acceptable for publication, you may indicate that here to bypass the “Comments to the Author” section, enter your conflict of interest statement in the “Confidential to Editor” section, and submit your "Accept" recommendation.

Reviewer #2: All comments have been addressed

2. Is the manuscript technically sound, and do the data support the conclusions?

Reviewer #2: Yes

3. Has the statistical analysis been performed appropriately and rigorously? 

Reviewer #2: Yes

4. Have the authors made all data underlying the findings in their manuscript fully available?

Reviewer #2: (No Response)

5. Is the manuscript presented in an intelligible fashion and written in standard English?

Reviewer #2: (No Response)

6. Review Comments to the Author

Reviewer #2: The introduction has been re-written and is more clear providing information requested such as explaining the meaning of IBWT

The initial comment has been addressed

Remove add or check if there is an additional technique missing on line 190

Line 260 add to between conform and research

I have attached my detailed review and apology for the delay

7. PLOS authors have the option to publish the peer review history of their article (what does this mean? ). If published, this will include your full peer review and any attached files.

**Do you want your identity to be public for this peer review?** For information about this choice, including consent withdrawal, please see our Privacy Policy .

Reviewer #2: No

---

## [Editor Report · Acceptance letter]

PONE-D-24-27908R1

PLOS ONE

Dear Dr. Ferreira,

I'm pleased to inform you that your manuscript has been deemed suitable for publication in PLOS ONE. Congratulations! Your manuscript is now being handed over to our production team.

Kind regards,

on behalf of

Dr. Mattias Gaglio

Academic Editor

PLOS ONE